# Next-Generation Influenza Vaccines and the Pandemic Horizon: Challenges, Innovations, and the Road Ahead

**DOI:** 10.3390/vaccines13111097

**Published:** 2025-10-27

**Authors:** Jessica Taaffe, Philipp Lambach, Pierre Gsell, Ioana Ghiga, Shoshanna Goldin

**Affiliations:** World Health Organization, 1211 Geneva, Switzerland; jessica.taaffe@gmail.com (J.T.); lambachp@who.int (P.L.); gsellp@who.int (P.G.); ghigai@who.int (I.G.)

**Keywords:** influenza vaccines, influenza vaccine R&D, vaccine development, next-generation influenza vaccines, pandemic preparedness

## Abstract

Background: Next-generation influenza vaccines have the potential to overcome the limitations of current seasonal influenza vaccines by providing more efficacious, broader, and longer-lasting protection and enhanced pandemic preparedness. However, their development is constrained by regulatory, financial, and scientific challenges. This study aimed to better understand these barriers and enablers from the perspective of vaccine developers. Methods: We employed a mixed-methods approach, collecting data through an online survey and follow-up interviews with 17 developers engaged in next-generation influenza vaccine R&D. Thematic analysis was used to identify key scientific, regulatory, and financial challenges, enablers and priorities for advancing development. Results: Developers reported a range of scientific and regulatory challenges, particularly the lack of established correlates of protection and uncertainty around evaluation criteria for novel platforms. High development costs and limited access to sustained funding were consistently cited as key barriers. Developers emphasized the need for clearer regulatory guidance, harmonized approval pathways, and validation of alternative immune markers. Collaborative approaches, including partnerships, consortia, and trial networks, emerged as critical enablers. Most respondents also reported leveraging their influenza vaccine R&D to support COVID-19 vaccine development, underscoring their relevance to broader pandemic preparedness. Conclusions: Next-generation influenza vaccines have the potential to significantly improve both seasonal influenza control and pandemic response. Realizing this potential requires coordinated action to address scientific, regulatory, and financial hurdles. Investment in regulatory innovation, sustainable financing, and collaborative R&D platforms will be essential to accelerate progress and ensure global access to improved influenza vaccines.

## 1. Introduction

Vaccination is the cornerstone of seasonal influenza prevention and control and the primary tool for influenza pandemic preparedness and response. Seasonal vaccines not only reduce influenza disease morbidity and mortality, especially in high-risk groups [1], but also offer the opportunity to establish and strengthen the infrastructure and platforms for pandemic response [2,3,4,5]. Current licensed seasonal influenza vaccines are safe and broadly used. Annual vaccination is recommended for high-risk groups to protect against severe illness. Vaccines may also be used to address outbreaks and exposure risk of zoonotic influenza, such as from A(H5) influenza, in the interpandemic period and emergence periods; vaccination will be a primary intervention during a pandemic [6].

Influenza vaccines have been available for more than 80 years, with the first licensed vaccine developed in the 1940s. Since then, vaccine technologies have evolved to improve their safety, immunogenicity, and manufacturing processes. Inactivated influenza virus vaccines have been formulated with adjuvant or higher antigen content to enhance immune responses in older adults. More recently, influenza vaccines developed using cell-based and recombinant technologies have become available. These technologies limit the risk of egg-adaptive mutations, which can affect antigenic match and impair vaccine effectiveness [7,8]. They also have potential for faster scale-up and reduce reliance on eggs for production, which is particularly important in a pandemic scenario. While influenza vaccines developed using mRNA technologies have not yet been licensed, they are anticipated to become available in the near future and have shown potential for superior efficacy when compared to traditional inactivated vaccines [9]. 

Despite the advancements, seasonal influenza vaccines still have certain limitations. Vaccine effectiveness varies widely, from less than 20% to around 60% [10,11,12,13,14,15,16,17,18,19,20,21,22,23,24], influenced by factors such as antigenic drift in circulating influenza viruses, vaccine type, and pre-existing immunity [25,26,27,28,29,30]. Influenza A(H3N2) virus antigens, in particular, tend to have lower vaccine performance, with effectiveness varying between different phylogenetic subclusters or variants within a single season [26,27,28]. Current seasonal influenza vaccines provide limited duration of protection and require annual administration. Vaccine strain selection occurs months before the influenza season, and production relies on processes that limit responsiveness to emerging strains. These limitations contribute to potential suboptimal matches and may come with decreased effectiveness.

Seasonal influenza vaccine production provides the foundation for pandemic influenza vaccine supply. The timeline for production of the first doses of a pandemic influenza vaccine with current technology is estimated between 4 and 6 months [31], though production could be faster with recombinant technology and/or synthetic-based manufacturing (mRNA vaccines). A meta-analysis showed vaccine effectiveness from the monovalent vaccine as high as 73% during the 2009 A(H1N1) pandemic [32]. In addition, a number of pre-pandemic (also referred to as zoonotic) influenza vaccines are licensed for human use against A(H5) [33]. There has been limited global use of pre-pandemic and zoonotic influenza vaccines, which limits the availability of data on the effectiveness and duration of protection of these non-seasonal influenza vaccines [6].

Overall, seasonal influenza vaccines’ moderate and variable effectiveness by season and target group, requirements for annual production and administration, and limited duration of protection have programmatic and cost implications. They are available in all six World Health Organization (WHO) regions and across all World Bank income classifications, although there is a gradient effect (e.g., correlation between national income level and the likelihood of having an influenza vaccination program) [34]. In 2022, influenza vaccines were available in 143 countries (74% of World Health Organization Member States) [35]. With the World Health Assembly Resolution 56.19, WHO Member States set a target of 75% influenza vaccination coverage for older adults and people with chronic conditions [36]. In 2024, while only twelve Member States reached this target for older adults and people with chronic conditions, 33 Member States (17%) achieved 75% vaccination coverage for at least one influenza vaccination target group [37].

Improved and next-generation influenza vaccines may address barriers to introduction/expansion of influenza vaccination programs, enabling countries to improve coverage rates in line with the World Health Assembly target for 75% coverage. In particular, next-generation vaccines may enhance the efficacy of vaccines, reduce the frequency of re-vaccination, and allow for vaccines to be used across multiple seasons. Depending on their breadth of protection, they may even provide some level of protection from a pandemic virus until the pandemic monovalent vaccine is able to be produced. A recent analysis estimates that next-generation influenza vaccines could avert 2.63–4.04 million hospitalizations and 519,000–826,000 deaths globally (compared to 1.85 million hospitalizations and 357,000 deaths from current seasonal influenza vaccines) by vaccinating all children under age 18 years. Even at higher prices than current influenza vaccines, next-generation vaccines could be cost-effective in many countries, especially if tiered-pricing was used for lower income countries [38].

Global health stakeholders, including WHO, have emphasized the urgent need for improved influenza vaccines. The WHO Global Influenza Strategy 2019–2030 prioritizes the need for next-generation influenza vaccines with improved breadth and duration of protection, enhanced effectiveness against severe disease, and reduced production time [39]. Multiple governments and organizations, including the United States [40], European Commission’s EU4Health program [41,42] the Bill & Melinda Gates Foundation [43], and the Sabin-Aspen Vaccine & Policy Group [44], have emphasized the need for next-generation influenza vaccines. In response, the influenza vaccine research and development (R&D) landscape has been supported by numerous global initiatives supporting innovation in this space [43,45,46,47,48,49,50] and a robust and diverse vaccine candidate pipeline, including pandemic products [33,51]. WHO will be publishing the updated Preferred Product Characteristics for Next-Generation Influenza Vaccines shortly. This work is complemented by the WHO R&D Blueprint and the Full Value of Improved Influenza Vaccine Assessment (FVIVA) [52]. The WHO R&D Blueprint functions as a global platform for collaboration to expedite R&D of medical countermeasures to respond to epidemics and pandemics. Influenza A(H5) is included within the list of priority pathogens for medical countermeasure R&D [53]. FVIVA is a structured WHO-led effort to assess the potential public health and economic impact of improved influenza vaccines through multi-dimensional evidence [54]. A full understanding of the R&D landscape for improved influenza vaccines, including enablers and challenges within this space, was conducted as part of the FVIVA project. This article fits within the FVIVA project by identifying enablers and challenges to next-generation influenza vaccine development.

In this manuscript, we provide an analysis of factors that can enable or impede research and product development on next-generation and universal influenza vaccines.

## 2. Methods

This analysis employed a mixed-methods approach combining quantitative and qualitative data collection. Data were gathered through an online survey and follow-up virtual interviews with representatives from organizations actively engaged in the development of next-generation influenza vaccines. The survey was designed to capture a broad range of information, including organizational characteristics, stages of product development, perceived challenges, and enabling factors within the current R&D environment. Semi-structured interviews were subsequently conducted to explore these themes in greater depth, allowing for the collection of nuanced insights and context-specific perspectives.

The study protocol (ERC.0004105) went through and was approved by the WHO Ethics Review Committee (ERC) process, which included review by external scientific experts.

### 2.1. Inclusion Criteria

Developers of next-generation influenza vaccines were identified through desk review, using the CIDRAP Universal Influenza Vaccine Technology Landscape [55] and information from a recent review of next-generation influenza vaccine pipeline [51]. Based on available contact information and a focus on developers with products in clinical development, at least 33 developers were invited to participate in the study and complete the online survey (some invitations were sent to lead consortium or funder points of contact, who then coordinated the responses from individual vaccine developers). Seventeen developers completed the online survey, and four of them participated in the virtual interview. The number of interviews was limited due to both participant availability and the need to balance depth of insight with feasibility. Despite the small number, interviews provided rich qualitative data. Thematic saturation was observed across key areas of interest, supporting the validity of the findings.

### 2.2. Online Survey and Virtual Interview

Development of the online survey and virtual interview guide were informed by a review of influenza vaccine and general vaccine literature as well as expert and other stakeholder feedback (e.g., FVIVA Technical Advisory Group members and external experts with similar credentials). The survey and interview included questions and discussion on individual products, different vaccine platforms, enablers or challenges within the preclinical and clinical development space, and other questions pertinent to improved influenza vaccine development (see Appendix A for survey and interview questions). The vaccine and influenza experts that reviewed this study’s protocol (see Acknowledgements) provided specific feedback on the survey and interview questions.

Responses to the survey were collected anonymously online through WHO DataForm software (DataForm v3) between July and September 2024 and were exported into Microsoft Excel for analysis. Interviews with developers who completed the survey were conducted via Microsoft Teams between November and December 2024, using a discussion-style format guided by a standard set of questions.

Prior to participation, all survey and interview participants were provided with detailed information about the study. Informed consent was obtained in accordance with ethical research practices.

Interviews were recorded with the consent of participants; transcripts were automatically generated using Microsoft Teams. Participants were given the opportunity to review their transcripts to verify accuracy and to approve the inclusion of information for reporting purposes.

## 3. Results

### 3.1. Overview of Developer Respondents and Their Vaccine Candidates in Development

Seventeen next-generation influenza vaccine developers completed the survey, providing responses and perspectives from developing a total of 51 vaccine candidates (38 in preclinical development, 13 in clinical development); nearly half of survey respondents reported having candidates actively in the clinical phase of development. Respondents reported as developers in biotech or academia; no developer from a major vaccine manufacturer provided responses to the survey. Developers reported using a wide range of vaccine platforms for their next-generation influenza vaccine development, including viral vector, recombinant protein, influenza-virus based, virus-like particle (VLP), non-VLP nanoparticle, mRNA or self-amplifying RNA (saRNA), and DNA platforms (Table 1).

Key themes that emerged from survey results and discussions during interviews were related to clinical evaluation and correlates of protection, regulatory pathways and guidance, funding and cost of development, collaborations and partnerships, and other scientific or technical issues (Figure 1). These topics are discussed in detail in the results below.

### 3.2. Clinical Evaluation of Next-Generation Influenza Vaccines and Correlates of Protection

Discussion with developers emphasized the importance of preclinical and early clinical studies to identify the appropriate measurements for clinical evaluation of novel vaccines, especially since performing certain assays on all subjects in a field efficacy trial can be technically, financially, and logistically challenging. Many of the vaccines in development are not designed to be protective through hemagglutinin antibodies (the approach of traditional influenza vaccines) and therefore cannot be evaluated with the same measures and assays as currently licensed vaccines. Developers cited one of the top challenges associated with the vaccine platform(s) they are using to be that the “immune response profile requires different measures of efficacy/immunogenicity than currently licensed vaccines” (Figure 2). Similarly, open-ended responses about preclinical development enablers and challenges indicated that the influenza hemagglutination inhibition (HAI) antibody titer, which is used to understand the protective effects of vaccination, could not be measured with their vaccine candidates. Correlates of protection are needed for next-generation vaccines. Resolution of this issue, such as through the “identification of surrogates of protection beyond HAI titers,” could enable more streamlined movement through preclinical into clinical testing. In addition, if a correlate of protection was identified, it could be used to facilitate bridging studies for different age groups or populations, offering a more efficient approach than field efficacy studies for each indication.

A correlate of protection will need to be established for a new vaccine and used for its approval and/or a large Phase 3 efficacy study will be required to demonstrate protection. Developers are skeptical of the former scenario happening before and without the latter. Even then, challenges are expected with establishing and using a new correlate for approval, requiring significant stakeholder support and advocacy. One developer indicated that scientific leadership from global health organizations are needed to establish that “the science is good; a new correlate is possible. Let someone use it from the standpoint of getting approval and then after approval, you could do more efficacy surveillance.” The same developer noted that human challenge studies could be used to identify and establish new correlates of protection, which would be an innovative regulatory approach, if allowed.

Developers highlighted the need for clear guidance on the value of alternative outcome measures for licensure, such as human challenge studies and alternative study models. This issue around evaluation of performance and efficacy is critical for clinical trials. Specific guidance on requirements for approval as part of the regulatory process was cited as a top enabler by all developers with products actively in clinical development. Inversely, stringent regulatory requirements were indicated as a top factor that would hinder clinical development, as selected by all 8 developers with products in clinical development (Figure 3). Developers articulated the value of consensus and alignment between different regulatory bodies on required evaluation measures and methods of analysis.). Developers articulated the value of consensus and alignment between different regulatory bodies on required evaluation measures and methods of analysis.

As a goal of next-generation influenza vaccine development is broader and longer duration of protection, how to appropriately measure these aspects is an additional challenge. A traditional approach to this, as described by one developer, may be that developers seek initial licensure based on one year of data. As data become available, the developers may use post-licensure follow-up to measure longer-term or broader protection. Indeed, one developer pointed out that the earlier they could achieve licensure, even conditional approval for a pre-pandemic or seasonal indication, the sooner they could use post-marketing studies to measure effectiveness against antigenically divergent strains that arise over influenza seasons. Other developers also described this post-licensure approach to evaluating longer-term or broader protection. However, prior to demonstrating multi-year breadth or duration of protection data, another developer indicated that large scale usage, particularly in some countries, may be limited with a new vaccine.

### 3.3. Regulatory Pathways, Guidance, and Processes

Navigating regulatory processes and uncertainty around requirements for regulatory approval of novel influenza vaccines was another key theme that emerged in this study. Clear and efficient regulatory pathways with specific guidance on requirements for approval, including for new influenza vaccine platforms and technology was the top selected factor to assist clinical development, whereas stringent regulatory requirements or delays in the regulatory approval process was among the top responses from developers on factors that would hinder clinical development (Figure 3).

In open-ended responses focused on preclinical development, expedited regulatory review and Investigational New Drug (IND) packages, the administrative, scientific, and safety documents required by the US Food and Drug Administration before testing a new drug or biological product in humans, were additionally identified as supportive to the development process (Table 2). Nearly half (8/17) of developers indicated that they had engaged with a national regulatory agency, specifically North American and European regulatory bodies, and most developers interviewed planned to launch their product(s) first in the United States of America or Europe.

Some of the developers began discussing the regulatory pathway with regulators while their products were in the preclinical development phase. In general, regulatory engagement experiences were mixed, differing between developers and regulatory agencies. Some agencies were considered rigid in their regulatory pathways and requirements and slow to respond to minor protocol changes that could delay studies. One developer also noted that the “ability to take risk-based assessments and to participate in scientific advice meetings” was very helpful in their engagement with regulatory agencies and another indicated that they would “like to have a regulatory environment where small trivial choices could be responded to much more quickly”. Other agencies were described as easier to work with, quicker due to more experience with clinical trials, and more flexible in their evaluation measures and approval pathways. Such characteristics influenced developer decisions on where to conduct clinical trials and seek regulatory approval. Developers explained that where feasible, they intended to conduct a “first in humans study” in a country with experienced regulators and and later conduct phase studies in other countries. They also indicated the preference to seek initial licensure with a rigorous but also flexible agency which would facilitate licensure in other countries.

In addition to specific guidance and requirements from regulatory bodies, developers expressed interest in having more clarity from other vaccine development guidance, like funder Target Product Profiles (TPP), Strategic Plans, WHO’s Preferred Product Characteristics, etc. One developer indicated that such documents were useful but noted that they often seemed like more of a “wish list” versus a list of characteristics critical for approval. Instead, developers indicated that what would be most useful in terms of vaccine development guidance would be consensus and synergy among from regulatory bodies on what is needed to receive approval for next-generation influenza vaccines. This guidance should include requirements for demonstration of protection as well as outlining the appropriate regulatory pathways, study design, and evidence needed for approval. This synergy between regulatory bodies could facilitate a more streamlined and resource-efficient development process, reducing regulatory administrative challenges associated with developing and licensing vaccines, such as preparing multiple dossiers for the same vaccine to receive approval in different countries.

Vaccine performance and delivery improvements provided by next-generation influenza vaccines could serve as enablers for regulatory approval, overcoming discussed barriers. For example, one developer noted that a universal vaccine demonstrating immediate mucosal immunity in the nose and blocking transmission would facilitate an easier pathway to licensure. Another described how demand for a vaccine with delivery improvements (e.g., oral capsule, microarray patches, inhaled vaccines) might enable approval through a new pathway: “If we could convince a regulatory agency that [our vaccine’s correlate of protection] was good, we might be able to go do a Phase Three trial, measure the correlate and then get approval… there may be places where [an easily administered vaccine] that has good efficacy (as good or better than standard of care) would make a huge difference, so approval could be done by a new correlate.”

### 3.4. Funding and Cost of Development

High costs of clinical trials and the funding required to support development of next-generation influenza vaccines were identified as significant challenges for developers. Cost of development was the second-most cited challenge with using next-generation influenza vaccine platforms (Figure 2). Developers selected funding, whether “adequate, predictable, and sustained” (two-thirds of responses) as assisting clinical development while limited and poorly sustained funding was a barrier (Figure 3). Funding for and cost of preclinical development is also a key issue.

The cost of development is related to both the vaccine performance evaluation and regulatory issues also cited. Developers explained that next-generation influenza vaccines will likely require large-scale trials to demonstrate vaccine efficacy or effectiveness in the absence of accepted correlates of protection. The high number of study participants required to do so for Phase 3 trials will increase the overall clinical development costs. In addition, the developers cited the costs related to scaling up commercial manufacturing through in-house production or with a contract manufacturing organization to enable clinical trials to be conducted. One developer estimated that a trial with an established correlate of protection might require 3000 participants while a trial that needs to demonstrate efficacy from a clinical endpoint could require 10,000–20,000 participants. Costs would increase for next-generation influenza vaccines seeking to demonstrate longer duration or broader protection, as these would require multi-year follow-up studies. Discussion with developers indicated a need for guidance on what should be evaluated for approval (including appropriate assays for evaluation) to enable careful planning of advanced stage trials and optimal usage of resources.

In addition to high costs of advanced clinical trials, developers also cited the costs associated with Good Manufacturing Practice (GMP), using contract research organizations (CROs), and conducting large animal studies (like non-human primates) in their open-ended responses about the preclinical stage of development (Table 2). Manufacturing costs were identified as a top challenge with using next-generation influenza vaccine platforms (Figure 2). Mitigation of costs through risk sharing across development partners was cited as a benefit of product development partnerships or consortiums. One developer indicated that they were used to working with academic trial networks, which are considerably less expensive than CROs.

As a result of the high cost of development and product commercialization, developers noted that partnership with a major vaccine manufacturer would be necessary at some point in development. “For us to be successful, we’re going to have to partner with a major vaccine company. We can’t commercialize [our vaccine] ourselves.” A potential conflict is if that manufacturer also has their own influenza vaccine(s) on the market: “If they were interested in developing or commercializing our vaccine, they would be potentially displacing a huge investment in their own vaccines.” Another developer emphasized this challenge and concern: “Small pharma can’t afford costly Phase 3 efficacy trials and requires partnering with big pharma who are reluctant to change the current influenza vaccine business model.”

Securing the necessary funding to support development of new vaccines through clinical development was indicated as a major challenge. Developers noted earlier phase studies cost considerably less than advanced trials, making it easier obtain funding for them. However, even raising the funds for early clinical development can be difficult, as developers may not have the advanced or type of data that investors or funders may seek to fund continued development of their vaccine candidate. For example, one developer expressed frustration over one grant’s technology readiness requirement: “You can’t get the funding to get to the technology readiness level that qualifies you for the grant you really need.”

There are financial risks in developing next-generation influenza vaccines, primarily because there is uncertainty over the level of improvements in performance (breadth, duration, efficacy) that can be realistically made and what the demand and market will be for those vaccines. Even for a successful product, returns on vaccine development investment likely will not be seen until several years after it has been on the market. One developer notes that, “You need to have some faith that you’re going to be able to get your product on the market and recoup those [development] costs.”

### 3.5. Collaboration and Partnerships

Collaboration and partnerships are enablers of clinical development (Figure 3). Ten of the developers indicated that they had explored or engaged in a product development partnership or other agreement (ex. technology transfer) (Table 3). Specific examples included partnerships with influenza vaccine producers, working with contract development and manufacturing organization and CROs for manufacturing processes and initial proof of concept data, including GMP material production, and scientific collaborations to probe vaccine mechanisms of action or large-scale research and co-development through consortiums. This was also reflected in the open-ended responses on preclinical development which highlighted the value of “complementary expertise all the way from basic science to vaccine development to manufacturing and device expertise” to support development. One developer credited experienced clinical trial partners with enabling efficient and effective engagement with regulatory authorities, and another indicated that technical guidance from their funder was very useful.

### 3.6. Other Scientific or Technical Issues

Next-generation influenza vaccine developers also noted an array of additional scientific or technical issues in developing their vaccine candidates.

Vaccine technology considerations feature prominently in the development process, as developers noted the adaptability of vaccine technology and other innovations as enabling preclinical development, for example, technology supporting multiple and cross-pathogen antigen inclusion or usage with other licensed vaccines. Indeed, “adaptability for different antigens in the platform” and “flexibility in formulation and valence” were among the top favorable characteristics associated with using next-generation influenza vaccine platforms (Figure 2). Features related to product performance, such as safety, immunogenicity strength, breadth, and type of response during preclinical development were also cited as favorable characteristics associated with next-generation vaccine platforms (Figure 2) and enabling in the preclinical development phase.

Developers also recognized the importance of programmatic suitability, citing favorable characteristics such as ease of administration and cold chain or storage considerations related to their next-generation product platforms (Figure 2). One developer emphasized that focusing just on product development costs is limiting, as improvements in the vaccine products that translate into delivery and distribution efficiencies (eg., self-administration, shelf-stability) could reduce vaccine program costs. Another developer noted that a vaccine with broader or longer protection could reduce seasonal administration logistics and costs and strengthen pandemic preparedness. For example, staggered immunizations with a vaccine that was effective for three years could enable programs to focus on reaching a third of the recommended population each year. This could support countries to increase programmatic efficiency and reach higher coverage rates.

Ease of production and scalability, manufacturing challenges and cost, and general expertise in manufacturing processes were associated with next-generation platforms and key considerations in preclinical development (Figure 2, Table 2). Developers also noted that the availability or lack of suitable animal models and reagents, including viruses for vaccine performance evaluation, and preclinical results not being predictive of human responses were technical enablers/challenges for clinical and preclinical development. One developer explained that using the ferret model, a gold standard animal model for influenza, is challenging for their vaccine, which relies on cell-based immunity for protection. Reagents to characterize T-cell responses in ferrets are limited, and this model is more suited for evaluating antibody-based vaccines. Therefore, it is very difficult to correlate any immune response to protection observed in this species.

A key theme that emerged when interviewing developers about their development process was the primary focus on strong science and letting that guide development versus designing and developing a product to meet specific criteria or a public health need, such as those outlined in TPPs, at least early in development. One developer said, “I’m interested in defining what we can achieve and then working on what we could do and how could we best use that vaccine.” While acknowledging the importance of the greater public heath context of vaccine development and vaccination programs, another developer emphasized their role bringing the science and product forward. At a later stage in development, addressing public health or programmatic aspects could be considered.

We also investigated the role that next-generation influenza vaccine development played in supporting COVID-19 vaccine development. Almost all (13 of 17) developers that responded to the survey reported leveraging existing technical expertise, development efforts, and production capabilities from their influenza vaccine R&D programs to support COVID-19 vaccine development (Table 3). Several next-generation influenza vaccine platforms were leveraged for use against COVID-19 or adapted for combination influenza–COVID-19 vaccines, with some even receiving licensure for COVID-19. In turn, safety data obtained post-licensure from previous and ongoing studies with one licensed COVID-19 vaccine are providing valuable information to improve development of an influenza vaccine using similar technology. Clinical data generated from influenza studies helped inform COVID-19 vaccine development, while manufacturing processes, immunological assays, and correlate modeling approaches were adapted to meet the needs of the new indication. In addition, preclinical work, including animal models and the use of respiratory virus reagents, provided a foundation for accelerated COVID-19 vaccine development.

### 3.7. Top Needs for Clinical Development Progression

Developers with products actively in clinical development (*n* = 8) were asked to list their top three needs to advance their product to the next stage of clinical development or product approval, for those in Phase 3 clinical trials, aside from promising clinical trial results. Their open-ended responses reflected the same issues identified from other questions in the survey, with regulatory and funding needs featured prominently. Funding was the most frequent response and includes financial resources to support GMP manufacturing at large scale and assay validation for trials. It also includes funding to support large scale efficacy, challenge, and proof of concept to generate correlates of protection data studies. Responses from developers during both the survey and subsequent interviews highlighted the critical role of regulatory guidance and approval and standardization within the field—the need for well-defined endpoints for clinical studies, standardized between trials, was cited, as well as standardization of T-cell correlates of protection. Additionally, developers included the need for “a clear regulatory path for universal influenza vaccines” and “regulatory guidelines for immunogenicity-based approvals for novel influenza vaccine formulations”. One developer indicated that regulatory approval was needed to move to the next stage of clinical trials.

Other needs were technical, scientific, or intellectual property related. Manufacturing scale-up and process optimization was required by some developers before moving forward, and the need to understand the vaccine mechanism of action and licensing agreements were also cited.

## 4. Discussion

Our study revealed significant and interconnected scientific, regulatory, and financial challenges associated with the development of next-generation influenza vaccines. It also highlighted factors and approaches that have enabled movement forward or could do so for these vaccines.

The identification and acceptance of new correlates of protection remains an essential, yet unresolved, issue. Developers underscored the need for robust preclinical and early clinical studies to define relevant immune markers for novel platforms. Emerging candidates often induce immune responses, such as T-cell activation or mucosal IgA, that are not captured by traditional serological assays. Identification of which measures should be used as a correlate of protection will likely be vaccine or platform-specific and will require scientific discussions involving developers, vaccine experts, and regulators alike. There is an urgent need for guidance on how new correlates of protection can be validated and accepted to streamline future evaluations and reduce reliance on large-scale efficacy trials, particularly when expanding indications to additional populations after initial licensure. Key meetings over the past two decades have explored these issues, summarizing research progress and discussing potential approaches for establishment and use of correlates of protection in regulatory pathways [56,57,58]. Movement forward, at this point, will require dedicated and collective action between developers, academics, and regulatory agencies to establish consensus and publish initial guidance.

Developers emphasized the need for clearer regulatory guidance, including critical requirements for next-generation influenza vaccine approval and alternative approaches for evaluating vaccine performance. Many next-generation candidates diverge from traditional hemagglutinin-based immune responses and adjuvant use may affect immunogenicity profiles [59,60,61,62,63], necessitating consideration of alternative immunogenicity metrics, corresponding correlates of protection, and innovative study designs like human challenge models to investigate them. An abundance of vaccines with potential for broader or longer duration of protection are currently in development [51]. These include candidates utilizing mRNA platforms, recombinant protein subunits, and novel adjuvants aimed at enhancing cross-protective immunity against diverse influenza strains. Specific guidance is necessary on how broader and/or multi-season protection should be evaluated and pathways for approval. Flexible and risk-based regulatory approaches, as well as harmonized guidance from regulatory bodies could facilitate licensure in multiple markets while maintaining rigorous standards.

High costs of clinical development coupled with challenges in obtaining funding are constraining progress. Developers reported difficulties securing sustained and sufficient investment, especially for Phase 2 and 3 clinical trials where costs escalate dramatically. Taking a product through commercialization is difficult for academic and small biotech developers without appropriate investment or partners (e.g., large pharmaceutical companies). This funding and investment challenge is compounded by uncertainties around regulatory requirements, future funding, and potential market demand. An unsupportive or uncertain funding environment could significantly hinder next-generation influenza vaccine development and stall the progress that has already been made. Without sufficient financial support, investment in these products may diminish, development programs may be deprioritized, and academic or small biotech research institutions may abandon projects. Bridging the so-called “valley of death” between early-stage research and late-stage clinical development remains a significant challenge, particularly in the current global funding context.

Funding awards appropriate for all phases of development are essential for next-generation influenza vaccine development, especially amounts that can support large-scale efficacy trials. Targeted financial support, for example, for activities like manufacturing scale-up, clinical assay validation, or evaluating mechanism of protection, could augment existing funding streams and give development programs the boost needed to generate key data, strengthen their value proposition, and increase their appeal to investors and major funders. Public–private partnerships and innovative financing mechanisms, such as milestone-based funding models or advanced market commitments, could further incentivize development.

Collaborative approaches emerged as an important enabler. Product development partnerships, consortia, and academic trial networks not only distribute costs and reduce financial risk but also provide access to critical scientific, regulatory, and other technical expertise. These collaborations are especially valuable for smaller developers navigating the complex landscape of vaccine development. Examples of collaborative initiatives that have supported this field include the US National Institute of Allergy and Infectious Diseases’ Collaborative Influenza Vaccine Innovation Centers (CIVICs) program [45], the European Commission funded Indo-European Consortium for Next Generation Influenza Vaccine Innovation (INCENTIVE) [46], and Flu Lab [47]. Continued investment in these large-scale collaborative approaches for vaccine development is critical to drive innovative products forward. Efforts to foster regional manufacturing and research capabilities, such as the WHO mRNA Technology Transfer Program which supports next-generation vaccine development in low- and middle-income countries (LMICs), are also enhancing global vaccine equity and pandemic preparedness.

Other stakeholders and partners have supported next-generation influenza vaccine development through agenda setting and prioritization of activities within the space. Historically, the US government has been a champion of next-generation influenza vaccine development for both funding and provision of technical expertise. However, the US government canceled contracts for the development of next-generation and universal influenza vaccines in early 2025. Other key partners that provide technical expertise, norms and standards setting, and coordination functions for the development of next-generation-influenza vaccines include WHO, the European Medicines Agency, the Wellcome Trust, the Global Funders Consortium for Universal Influenza Vaccine Development, and CIDRAP. The Influenza Vaccines Research and Development Roadmap (IVR) [48], a strategic planning tool for developing improved influenza vaccines, is a key product of this global coordination and engagement. In addition to scientific challenges and knowledge gaps, the Roadmap highlights critical policy, financing, and regulatory barriers that hinder progress in vaccine development. Many of these obstacles, identified through input from over 100 stakeholders across 29 countries, are echoed in the experiences reported by developers in our study. Similarly, the Sabin-Aspen Vaccine Science & Policy Group has underscored the same scientific, financial, and regulatory challenges [44], reinforcing the broader relevance and validity of our findings in this area.

A key finding in our study was the linkage between next-generation influenza vaccine development and COVID-19 vaccine development. Nearly all developers reported leveraging aspects of their influenza vaccine R&D programs to accelerate COVID-19 vaccine development, highlighting the cross-applicability of technological and scientific investments. The success of several licensed COVID-19 vaccines, many based on the same platforms being used for next-generation influenza candidates, underscores the relevance of influenza vaccine innovation to broader pandemic preparedness. For example, the global use of mRNA vaccines during the COVID-19 pandemic familiarized regulatory agencies with the process of approving vaccines developed with this platform. In addition, countries gained experience deploying next-generation vaccines (e.g., developing ultra cold chain capacity, communicating about mRNA vaccine products, providing trainings for health workers to administer the vaccines, etc.). This experience may enable countries to more rapidly introduce next-generation influenza vaccines as they become available, particularly in a future pandemic context.

Improvements in novel influenza vaccine candidates include alternative administration approaches (e.g., microarray patches, oral capsules, or intranasal formulations, etc.) that could be game changers for epidemic and pandemic influenza vaccination. Vaccines that do not require syringes or cold chain would revolutionize countries’ ability to store, deploy, and administer vaccines. In addition, these vaccines could dramatically reduce the waste management needed for pandemic vaccination campaigns, a consistent issue during the 2009 (influenza) and COVID-19 pandemics.

Beyond technological parallels, influenza vaccines are a cornerstone of pandemic preparedness. Seasonal influenza vaccination programs establish critical infrastructure, including surveillance systems, immunization logistics, regulatory frameworks, and public trust, that can be leveraged during emerging outbreaks. Strengthening influenza vaccination programs therefore not only addresses current public health threats but also serves as a strategic investment in global health security.

The COVID-19 experience demonstrated that expedited regulatory pathways, flexible evaluation criteria, and real-time data integration are feasible and effective, offering valuable lessons for accelerating influenza vaccine development. Moving forward, efforts to develop vaccines for both pathogens are likely to remain mutually informative. The pursuit of pan-sarbecovirus vaccines mirrors the long-standing goal of broadly protective and universal influenza vaccines, with shared challenges in defining breadth of protection and selecting appropriate clinical endpoints.

While the results of this analysis are limited to and by the profile of the 17 developers that responded to the survey, it is worth noting that viewpoints were provided by developers of all technology platforms recently characterized in development for next-generation influenza vaccines [51]. Our study included eight developers with a product in clinical development, which represents one-third of those documented as of April 2024 [51]. We captured responses from 15 developers with preclinical products, but this is a much larger and rapidly changing space [55]. Although the sample size was modest, the consistency of themes across respondents and alignment with previously reported sector-wide challenges support the robustness and relevance of our findings, suggesting that they are likely representative of broader trends within the field.

Mostly notably, our results are limited to perspectives from developers in academia or small biotech, as we did not receive any responses from major pharmaceutical companies. While our anecdotal experience suggests that similar overarching challenges are acknowledged within the pharmaceutical sector, important differences in resources, technical expertise, and operational capacity likely shape how these challenges are encountered—a perspective not captured in our study. Perspectives from major manufacturers would undoubtedly provide valuable complementary insights—particularly regarding prioritization among their overall product pipeline, late-stage development, and commercialization—however, most current next-generation influenza vaccine R&D originates from academic and small biotech entities [51]. As such, our results capture the views of the groups most directly involved in early innovation, even if they do not encompass the full industrial spectrum. It should also be noted that for those survey responses from individual developers that were coordinated by a lead consortium or funder point of contact, this method of collection may have influenced the responses, potentially limiting responses or biasing responses in a way favorable to the point of contact. Future direct engagement with developers (including major vaccine manufacturers) will be valuable in validating this study’s results and measuring the progress made to address the key barriers over time.

## 5. Conclusions

Influenza continues to pose a significant global health threat, with vaccines remaining the most effective tool for preventing illness and reducing pandemic risk. While current influenza vaccines have been invaluable tools in reducing seasonal disease burden, next-generation vaccines offer a crucial expansion of our pandemic preparedness toolkit by fostering broader, more durable immunity that can anticipate viral shifts beyond the reach of traditional antibody-focused approaches. Rather than replacing established vaccines, these innovations complement and strengthen our defenses by engaging diverse immune pathways, such as mucosal and T-cell responses, that have been underutilized to date. Realizing their full potential hinges on adaptive regulatory frameworks and sustained support for innovation across the spectrum of developers.

Drawing on lessons from recent rapid-response vaccine platforms, the path forward calls for scientific, regulatory, and financial barriers to be addressed. Overcoming these challenges requires sustained collaboration among developers, regulators, funders, and global health stakeholders. Clearer regulatory guidance, harmonized evaluation criteria, and validated correlates of protection are essential to facilitate the development and licensure of novel vaccine platforms. At the same time, targeted and sustained funding across all stages of development—particularly for advanced clinical trials—will be critical to ensure promising candidates reach the market.

Greater investment in regional research, including developing capacity for implementation research, epidemiological and economic modeling, as well as studies assessing the cost-effectiveness and public health impact of vaccines is essential to ensure country ownership and use of context-specific evidence to inform introduction and sustained use of next-generation vaccines that is equitable and tailored to country needs. Strengthening regional manufacturing is equally important to ensure equitable access and build response capacity to future pandemics. It is also important to increase funding to build public communication capacities at national and sub-national levels to build trust in vaccines and thus ensure uptake, particularly when vaccines are developed with novel platforms that may create hesitancy among future vaccine recipients.

The future of next-generation influenza vaccines holds great promise. Continued investment, strengthened partnerships, and coordinated action across sectors are needed now to accelerate progress and fully realize their potential for seasonal protection and pandemic preparedness. Resilience and future-oriented Influenza vaccine ecosystems bear the promise to improve prevention of annual epidemics but also serve to protect against next pandemics.

## Figures and Tables

**Figure 1 vaccines-13-01097-f001:**
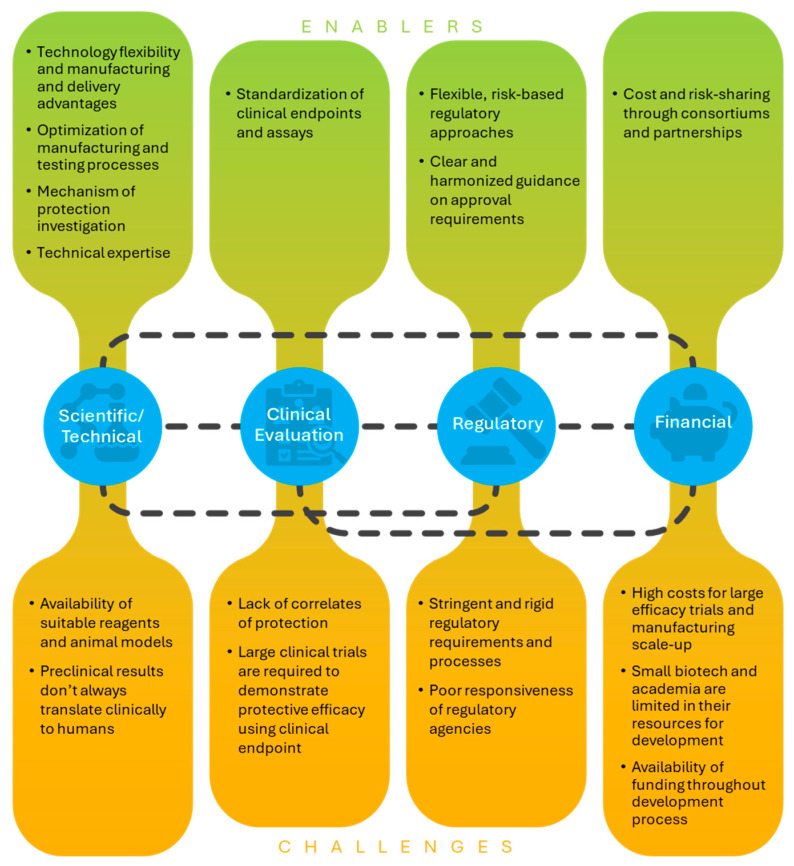
Key issues within next-generation influenza vaccine development. The main issues affecting next-generation influenza vaccine development are summarized, as described by developers. Enablers and challenges within the different categories that have affected or could affect the development process are included in green (enablers) or orange (challenges) containers. Identified issues within categories are interlinked, often influencing issues within other categories. Collaboration and partnerships were also identified as key enablers of development with examples within all categories.

**Figure 2 vaccines-13-01097-f002:**
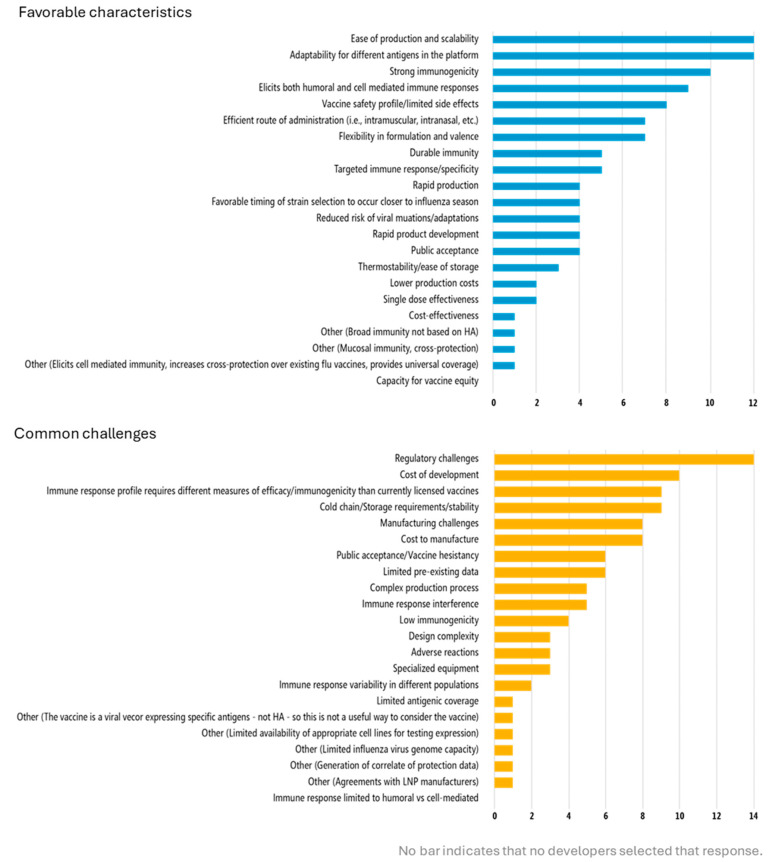
Favorable characteristics of and challenges with using next-generation influenza vaccine platforms. Developers were asked to select the top favorable characteristics and common challenges (maximum five each) associated with using their vaccine platform(s) for next-generation influenza vaccine development. The number of selected responses is shown for each option. If “Other” was selected, the open-ended response provided by the developer is included. Abbreviations: HA, hemagglutinin; flu, influenza; LNP, lipid nanoparticle.

**Figure 3 vaccines-13-01097-f003:**
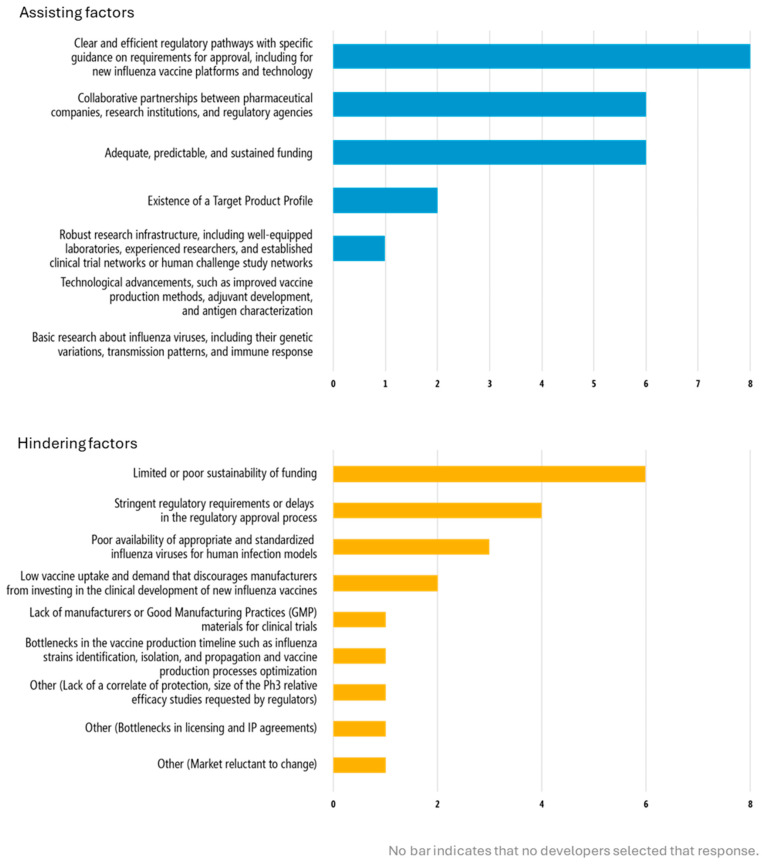
Factors that may assist or hinder clinical development of next-generation influenza vaccines. Developers with vaccines in clinical development (*n* = 8 developers) were asked to select the top factors that may assist or hinder (maximum 3 each) clinical development of their next-generation influenza vaccine(s). The number of selected responses is shown for each option. If “Other” was selected, the open-ended response provided by the developer is included. Abbreviations: Ph3, Phase 3; IP, intellectual properties.

**Table 1 vaccines-13-01097-t001:** Overview of developers that responded to survey.

Developers That Responded to Survey (Total)	17
**Developers with products in different stages of development**	
With vaccines in preclinical development	15
With vaccines in clinical development	8
**Developers using different vaccine platforms**	
Viral vector	3
Recombinant protein *	4
Influenza virus-based	6
VLP	3
Non-VLP nanoparticle	4
mRNA or saRNA	6
DNA	2

* Includes peptide-based approach.

**Table 2 vaccines-13-01097-t002:** Enablers and challenges associated with the preclinical development of next-generation influenza vaccines. Developers were asked to identify the top challenges and enablers (maximum three each) associated with the preclinical development of their next-generation influenza vaccine(s), focusing on issues that challenge(d) or enable(d) movement through preclinical testing and to clinical development. The open-ended responses are summarized and thematically categorized in the table.

Category	Enablers	Challenges
Product performance/ Evaluation	Safety Desired immune response or outcome, including in animal models, achieved	Lack of correlates of protection Desired immune response or outcome, including in animal models, not achieved Vaccine mechanism of protection not hemagglutinin inhibition based Preclinical data not predictive of human response
Funding or funder support/Cost of Development	Risk sharing across development partners Technical support	Cost of Good Manufacturing Practices, using contract research organizations, testing, including large animal studies like non-human primate studies
Technology/Manufacturing	Used successfully with other viruses or licensed vaccines Adaptability for antigen inclusion, including multiple antigens Manufacturing ease Innovation in platform or unique technology Easily tested in preclinical experiments	
Regulatory	Expedited regulatory review Investigational new drug (IND) package	Lack of clear regulatory guidance on value of alternate outcome measures for licensure (e.g., human challenge studies, humanized animal models, organoid data)
Research/Product development	Availability/suitability of animal models Availability of reagents (including viruses) Ability to test iterations Identification of appropriate immunogenicity measures and surrogates of protection beyond hemagglutinin inhibition Good laboratory practices for toxicology, pharmacology, and immunogenicity Expertise in manufacturing Collaboration to ensure expertise across all development stages Optimized process development Optimizing manufacturing and testing processes	Availability/suitability of animal models Availability of reagents (including viruses) Modified viruses less fit

**Table 3 vaccines-13-01097-t003:** Additional questions about partnerships/collaboration, regulatory agency engagement, and leveraging of expertise for COVID-19 vaccine development.

Question	Number of Yes Responses
Have any product development partnerships or other agreements (e.g., technology transfer) been explored for the vaccine(s) named?	10
Have you engaged with a national regulatory agency to discuss the regulatory requirements for any of the vaccine(s) named?	8
Did you leverage existing technical expertise, development efforts, and/or production capabilities from your influenza vaccine R&D program for COVID-19 vaccine development?	13

## Data Availability

Data is available upon request.

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
