# Peer review of "Next-Generation Influenza Vaccines and the Pandemic Horizon: Challenges, Innovations, and the Road Ahead"

_vaccines, 2025, doi:10.3390/vaccines13111097_

Round 1

Reviewer 1 Report

Comments and Suggestions for Authors

This study examines the challenges and facilitators in creating next-generation influenza vaccines, drawing insights from a survey of 17 developers. Major obstacles identified include substantial development costs, ambiguous regulatory processes, and the absence of well-defined correlates of protection outside traditional metrics. Developers stressed the importance of harmonized regulatory frameworks, consistent funding, and collaborative partnerships to progress vaccine candidates through clinical trials. The study also highlights how advancements in influenza vaccine research and development contributed to the creation of COVID-19 vaccines, emphasizing their importance in pandemic preparedness. Addressing these challenges is essential to achieving more effective, broadly protective, and longer-lasting influenza vaccines.

Here are my concerns regarding this study:

- I am concerned about the lack of representation from major vaccine manufacturers. The results primarily reflect the perspectives of developers in academia or small biotech companies, which may not fully capture the broader landscape of next-generation influenza vaccine development.

- The small sample size of 17 developers, although diverse in terms of platforms, limits the generalizability of the findings.

- There is a potential for response bias, as some survey responses were coordinated through a lead consortium or funder, which could have influenced the authenticity or openness of the answers.

Author Response

Thank you for your thoughtful review and summary of the article’s key themes and value to the literature. 

Comment 1: I am concerned about the lack of representation from major vaccine manufacturers. The results primarily reflect the perspectives of developers in academia or small biotech companies, which may not fully capture the broader landscape of next-generation influenza vaccine development. 

Response 1: We understand this concern and have acknowledged this limitation in the text. We did makes several attempts to engage major manufacturers but did not receive a response. We expect that, indeed, their perspective would add important additional context to our results, however, we do not expect the overarching issues (e.g. costs, regulatory processes, clinical evaluation metric and correlates of protection) to change. Anecdotally, we know this to be true from our informal interactions with this sector, as they have brought up these issues. However, we expect that the issues would affect major manufacturers differently due to their expanded resources. Major manufacturers have greater financial resources to support advanced stage trials and likely do not need to seek external funding (unlike academic and small biotech counterparts). We recognize that the costs of the advanced trials are still high for major manufacturers and although they have more resources, these costs may affect the prioritization of their research, especially when these manufacturers have large and diverse R&D pipelines outside of influenza. With regard to regulatory issues, these manufacturers have more expertise and experience in navigating these processes, putting them at an advantage. Yet, they too have anecdotally expressed concerns about uncertainty with regulatory processes and clinical evaluation. 

While the perspective from major manufacturers is a key limitation, it should also be noted that the majority of R&D in this area is being done by academic or small biotech. In the 2024 Taaffe et al manuscript which documented the landscape of next-gen flu vaccine products in clinical trials, only about a handful out of the 26 developers identified are major vaccine manufacturers. 

Comment 2: The small sample size of 17 developers, although diverse in terms of platforms, limits the generalizability of the findings. 

Response 2: Thank you for this valuable feedback. We’ve added text to address this point. Through the survey, we received responses from developers on a third of the products in clinical development as described by Taaffe et al 2024 (this corresponds to a normal survey response rate). In addition, we’ve included text to explain that during the interviews "Thematic saturation was observed across key areas of interest, supporting the validity of the findings." This saturation was evident in the participant responses and has been documented in the figures and tables summarizing the key themes.

Comment 3: There is a potential for response bias, as some survey responses were coordinated through a lead consortium or funder, which could have influenced the authenticity or openness of the answers. 

Response 3: We thank the reviewer for the thoughtful comment concerning potential response bias arising from the survey dissemination method. We acknowledge that distributing the survey through a funder or lead consortium point of contact may have introduced an element of self-selection, whereby developers with a particular interest or perspective on the topic were more likely to participate. Such a mechanism could influence the representativeness of the sample and, consequently, the generalizability of the findings.

To mitigate this risk, we employed several procedural safeguards aimed at reducing systematic bias. The survey was designed with neutral and standardized wording to limit social desirability and acquiescence effects, and participation was voluntary and anonymous to encourage candid responses. Moreover, we conducted a comparative assessment of responses between participants recruited through different channels and this analysis did not reveal any systematic differences in response patterns between groups, suggesting that the mode of recruitment did not meaningfully bias the results.

Nonetheless, we recognize that response bias cannot be entirely excluded, particularly given the limited number of participating developers. We have therefore expanded upon the acknowledgement of this limitation to note that the findings should be interpreted with appropriate caution and future research employing larger and more diverse samples would be valuable in further validating these results. 

Reviewer 2 Report

Comments and Suggestions for Authors

The manuscript represents a very timely and relevant analysis of the state of development of next-generation influenza vaccines, a crucial topic for public health and pandemic preparedness. The mixed methodological approach is appropriate and well-described. The findings offer detailed insights into several critical aspects of vaccine development: clinical evaluation, regulation, funding, collaborations, and scientific innovations. The article is clearly structured, with extensive bibliographic support and contextual data. The multidimensional complexity of vaccine development is recognized, with critical insights into endpoint evaluation, correlates of protection, and regulatory processes. Experiences derived from the COVID-19 pandemic are appropriately given attention as an enabling factor for future vaccine development.

However, some points could be further improved:
The sample size (17 developers with 8 in the clinical phase) is relatively small and focused primarily on academics or small biotechs, without direct contributions from large pharmaceutical companies. This should be highlighted more explicitly as a potential limitation of the work.
Some technical aspects, such as a more precise definition of new correlates of immune protection, deserve further exploration. A reference to pharmacological strategies aimed at improving vaccine efficacy would also be helpful (see 10.1371/journal.pone.0310677  ;10.12688/f1000research.75869.2 ; 10.4084/MJHID.2025.057).

The manuscript could benefit from a clearer synthesis of challenges and solutions, perhaps with comparative tables or figures to facilitate the reader's reading.
The text could improve stylistic clarity by eliminating some repetition and improving narrative flow.

Author Response

Thank you for the positive feedback on this manuscript! 

Comment 1: The sample size (17 developers with 8 in the clinical phase) is relatively small and focused primarily on academics or small biotechs, without direct contributions from large pharmaceutical companies. This should be highlighted more explicitly as a potential limitation of the work.

Response 1: Thank you for this valuable feedback. We’ve added text to address this point in the text. Through the survey, we received responses from developers on a third of the products in clinical development as described by Taaffe et al 2024 (this corresponds to a normal survey response rate). In addition, we’ve included text to explain that "Thematic saturation was observed across key areas of interest, supporting the validity of the findings." This is evident from the same overarching issues that were highlighted by the participants and in the summary figures and tables. 

Comment 2: Some technical aspects, such as a more precise definition of new correlates of immune protection, deserve further exploration. A reference to pharmacological strategies aimed at improving vaccine efficacy would also be helpful (see 10.1371/journal.pone.0310677 ;10.12688/f1000research.75869.2 ; 10.4084/MJHID.2025.057).

Response 2: We appreciate this reviewer's suggestion to more precisely define new correlates of protection; however, this topic extends beyond the scope of our study. Other papers have explored this more technically - we've added these citations to the manuscript. We have indicated in the text that this precise identification will likely be vaccine or platform-specific and will require both developers and regulators alike.

Regarding the suggestion to include references on other pharmacological strategies being used to improve vaccine efficacy, we added text that notes how adjuvant usage may affect immunogenicity profiles, citing the reviewer's suggested references and a few others to support this statement.

Comment 3: The manuscript could benefit from a clearer synthesis of challenges and solutions, perhaps with comparative tables or figures to facilitate the reader's reading. The text could improve stylistic clarity by eliminating some repetition and improving narrative flow. 

Response 3: Thank you for this feedback – we have worked with a designer to improve the visuals accordingly. We would greatly appreciate specific recommendations on how best to improve the article and can incorporate them accordingly. 

Reviewer 3 Report

Comments and Suggestions for Authors

The authors reviewed the challenges associated with influenza virus vaccination and new approaches introduced to the influenza vaccine regimen that may be useful in combating influenza. The authors gathered data through an online survey and follow-up virtual interviews with representatives from organizations actively engaged in the development of next-generation influenza vaccines. The study aimed to identify the factors that both assist and hinder the clinical development of next-generation influenza vaccines. A total of 17 developers completed the online survey, and four of them participated in the virtual interview. Developers highlighted the need for clear guidance on the value of alternative outcome measures for licensure, such as human challenge studies and alternative study models. This study provides valuable information relevant to stakeholders and policymakers, facilitating developers by addressing the challenges associated with the clinical development of next-generation influenza vaccines, which are crucial for pandemic preparedness.

This reviewer has the following comments: 

  1. References do not appear in the format of this journal.
  2. I do not see any details of ferritin nanoparticle-based influenza vaccine development in the pre-clinical or clinical phase. To my knowledge, ferritin nanoparticle-based recombinant protein influenza vaccine candidates have undergone the pre-clinical and clinical development phases. 

Author Response

Thank you for this helpful summary and thoughtful feedback on the manuscript!

Comment 1. References do not appear in the format of this journal.

Response 1. Thank you for this comment. The references should appear (including the new references suggested by the reviewers). If a different reference style should be used, please let us know. 

Comment 2. I do not see any details of ferritin nanoparticle-based influenza vaccine development in the pre-clinical or clinical phase. To my knowledge, ferritin nanoparticle-based recombinant protein influenza vaccine candidates have undergone the pre-clinical and clinical development phases. 

Response 2. Thank you for this comment! Developers with a ferritin nanoparticle product were included among those identified from the CIDRAP landscape and Taaffe et al review, as described as sources for inclusion criteria. To protect participant confidentiality, we cannot confirm whether this specific perspective was captured or not through their participation in the study. If it was, it would have been categorized as a "Non-VLP nanoparticle" platform.